# Predicting student and instructor e-readiness and promoting e-learning success in online EFL class during the COVID-19 pandemic: A case from China

Chunye Yang[1]*, Defeng Xu[2]

1 School of Foreign Languages, Wuhan Business University, Wuhan, People's Republic of China, 2 College of Civil Engineering, Hubei Urban Construction Vocational and Technological College, Wuhan, People's Republic of China

* 1058227292@qq.com

## Abstract

Since the emergence and subsequent spread of the COVID-19 pandemic, students and instructors have faced unprecedented challenges and have been forced to shift traditional face-to-face classes online. This study, based on the E-learning Success Model (ELSM), seeks to examine the e-readiness level of students/instructors, judge the impediments that students/instructors encountered in different phases—pre-course delivery, course delivery, and course completion phase of the online EFL class, search for valuable online learning elements, and recommend recommendations for promoting e-learning success in online EFL classes. The study sample consists of 5914 students and 1752 instructors. The results demonstrate that: (a) both the students' and instructors' e-readiness level were slightly lower than the ready level; (b) three valuable online learning elements were teacher presence, teacher-student interaction, and practicing problem-solving ability; (c) eight categories of impediments during different phases of the online EFL class were technical challenges, learning process, learning environments, self-control, health concern, learning materials, assignment, and learning effect and assessments; (d) seven types of recommendations for promoting e-learning success were: (1) students: infrastructure and technology, learning process, content, curriculum design, teacher skills, service, and assessment; and (2) instructors: infrastructure and technology, human resources, teaching quality, content and services, curriculum design, teacher skills, and assessment. Based on these findings, this study recommends that further studies with an action research approach should be conducted to examine whether the recommendations are effective. Institutions should take the initiative to overcome barriers to engage and stimulate students. The outcomes of this research have theoretical and practical implications for researchers and higher education institutions (HEIs). During unprecedented times such as pandemics, administrators and instructors will have insights into implementing emergency remote teaching.

**Data Availability Statement:** All data are within the paper, its Supporting Information files, and the file Questionnaire(ZIP).

**Funding:** This research work was supported by the grants: "Methodologies and Teaching Practice of Moral Education in College English" from the Program of Wuhan Institution of Education and Planning Research (Grant No. 2021C167); "Mechanism Construction for Promoting Studies in Application-oriented Institutes" from Wuhan Business University (Grant No. 2022Y16), "Discursive Construction of Chinese Literary Classics Translation in Global Communication" from Wuhan Business University (Grant No. 2021KY009). The funders had no role in study design, data collection and analysis, decision to publish, or preparation of the manuscript.

**Competing interests:** The authors have declared that no competing interests exist.

## Introduction

Electronic learning (e-learning) is a key innovation in the delivery of education in the twenty-first century [1]. The term e-learning is often used interchangeably with distance education or distance learning [2], and now with online education. The field of distance education has changed greatly in the past decades. What was once considered a special form of education using nontraditional delivery systems, is now becoming an important concept in mainstream education [3].

With three years' dedication to improve the online education quality after the outbreak of the COVID-19 pandemic, online platforms, online Higher Educational Institutions (HEI) courses/resources and disciplines have been greatly expanded. According to the Chinese Ministry of Education (MOE) [4], 1100 thousand courses have been launched online by 1080 thousand teachers in HEI by the end of 2020, the number of college students studying online totaled 3.5 billion, and 91% of universities/colleges have offered online courses. 93% of the students were satisfied with the courses and various resources offered by universities/colleges. By February 2nd, 2021, there have been 22 online platforms in China providing 24,000 online higher education institutions courses free of charge, covering 12 disciplines at undergraduate level and 18 disciplines at higher vocational education level [5]. In 2021, there were 3012 higher education institutions in China. These statistics support the notion that distance education is currently the fastest growing form of domestic and international education [2].

In the past decades, researchers have been developing a readiness scale for online learning. What's the student/instructor e-readiness level and how should it be gauged and assessed? This results in a large volume of e-readiness-related studies to measure student/instructor e-readiness level [1, 2, 6, 7] with diverse methodologies or models [8–10] from different countries and regions, which include Jordan [11]; Tanzania [12], Barbados [1], Wuhan City [5], United Arab Emirates [13], Vietnam [14], to name but a few.

Zou et al. [5] studied that readiness in Wuhan City was slightly below the ready level for this emergency migration online for English education. Online education is now a new normal in higher education, to comprehensively gauge the electronic-learning readiness (e-readiness) of online EFL class in China and offer corresponding measurements to engage students, it is necessary to re-examine the underlying students' e-readiness level for online learning from a broader perspective covering more provinces in China, construct and validate measures to promote e-learning success, and to engage students in online EFL class.

The term "EFL" refers to "English as a foreign language". English learning, with a vast number of non-English major undergraduates in China, is a compulsory yet traditional part of the courses for at least freshmen and sophomores to study in a traditional way in a face-to-face way in the classroom, yet sequent to the COVID-19 pandemic, it was moved online. Until now, this web-based course, with three-years' perfection and enhancement, witnessed great changes in various aspects. However, while online education was bringing flexibility and convenience, the sudden shift of education from physical to online learning due to the pandemic has brought upon new worries and burdens around the globe [15], and problems occurred: expensive data traffic, weak and feeble signal, less interaction between students and instructors/students, absent-minded in class.

Studies on e-readiness level focused mainly on the following perspectives: learning styles, learning outcomes, learning opportunities, learning benchmarks, learning environment, and technology acceptance. However, researches have not provided solutions for promoting e-learning success and engaging students in the online class. Based on the E-learning Success Model (ELSM) [1, 2], this study aims to fill this gap by investigating students'/ instructors' e-readiness level for online EFL class, seeking for online EFL learning impediments, figuring out

learning elements for promoting e-learning success, and offering recommendations for engaging students and promoting e-learning success in online EFL class.

The following parts demonstrate literature review, theoretical foundation, and methodologies (materials and methods, data acquisition, RQ1-4). After that, it comes to the results and discussion. Research questions explored in the study are as follows:

1. What's the student/instructor e-readiness level in online EFL class in inland China?

2. What are the impediments for online EFL class?

3. What are the learning elements for promoting e-learning success in online EFL class?

4. What are the recommendations for promoting e-learning success in online EFL class?

## Literature review

### Electronic learning success model (ELSM)

The term electronic learning (e-learning) is defined by the Instructional Technology Council as well as the National Center for Education Statistics as the process of extending learning or delivering instructional materials to remote sites via the Internet, intranet/extranet, audio, video, satellite broadcast, interactive TV, and CD-ROM [2].

E-learning success model (ELSM) is to guide the research on searching for success factors for engaging and motivating students in different phrases of online education. It originated from the initial model—information system success model (I/S success model) which was proposed in 1992 [9], covering six major dimensions: system quality, information quality, use, user satisfaction, individual impact and organizational impact. In 2003, the I/S success model was updated as: system quality, information quality, use, user satisfaction, net benefits [16]. With decades years of research contribution to this model, Holsapple CW and Lee-Post A [2], in 2006, refined and enriched I/S success model, which was categorized into system design, system delivery and system outcome. In 2016, Glenda H. E. Gay [1] adapted the I/S success model from Holsapple CW and Lee-Post A to assess online instructor e-learning readiness before, during and after course delivery (see Fig 1). This research framework used an adapted version of Glenda H. E. Gay's model to predict student/instructor e-readiness and promote e-learning success (see Fig 2) in online EFL class from a Chinese perspective.

In each of the three phases, e-readiness factors are accordingly designated to students/ instructors. Combined with the prior research deficiencies, more factors, such as technological support, motivation, and help-desk service, are covered in this adapted framework. Therefore, student/instructor e-readiness level will be assessed specifically, objectively and comprehensively.

The framework makes explicit the process approach to measuring and assessing success [2]. Different student/instructor e-readiness levels in each of the phases indicate the attainment of success of each phase and present the rough impediments of online education. Therefore, this study investigates student/instructor e-readiness levels in each of the phases, impediments for online education, and recommendations for engaging students and promoting e-learning success in online EFL class.

### Online studies during the COVID-19 pandemic

E-learning systems have become renowned tools worldwide [17] and has emerged as a powerful medium of learning particularly using Internet technologies [18]. The unified theory of acceptance and use of technology (UTAUT) model was applied and extended in Ahmad

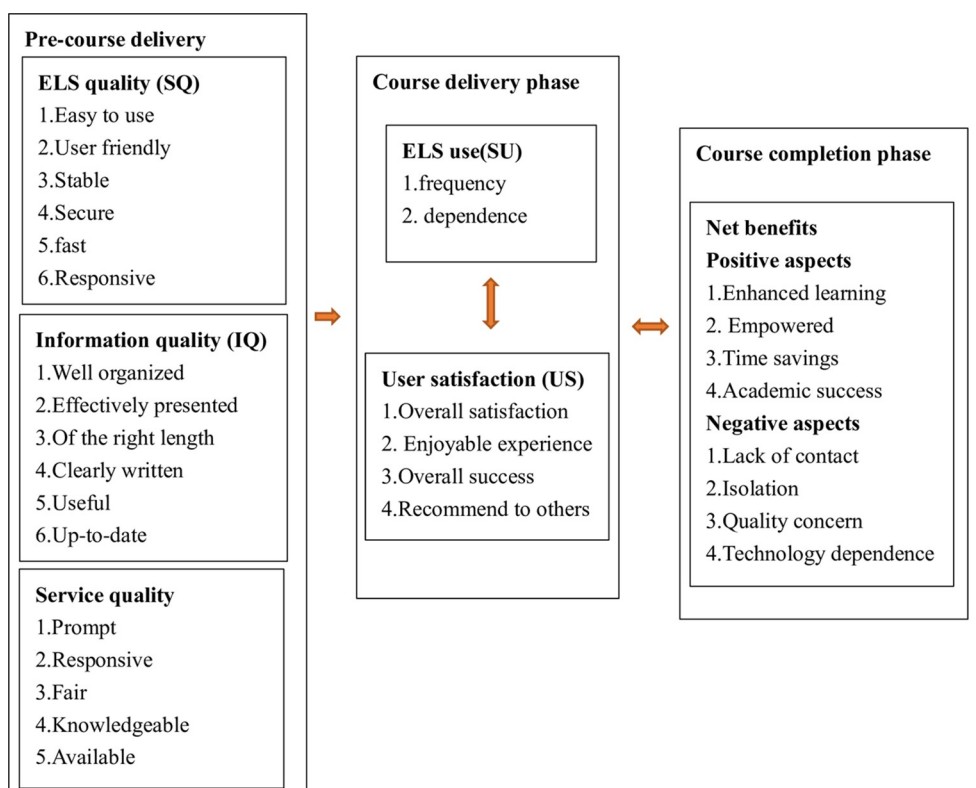

**Fig 1. Conceptual framework adapted from Holsapple and Lee-Post for online instructor e-readiness before, during and after course delivery (Glenda H. E. Gay, 2016).**

Samed Al-Adwan's study [19]. Learning tradition, self-directed learning, and e-learning self-ecacy were included in the newly added constructs [19]. These were also paramount elements for promoting online learning. Ahmad Samed Al-Adwan [20] investigated the factors that influence online students' continued usage intention toward e-learning systems and figured out that a context-specific factor that is a driver of successful implementations of e-learning systems. This broadened and deepened studies on searching for key elements on online learning.

In China, during the COVID-19 pandemic, studies on online English classes in the EFL context burgeoned. Wang Yongliang [21] figured out that boredom, known as an aversive, silent emotion, is still under-appreciated. Thus, Wang Yongliang [22] investigated the effects of aversive feelings like teacher boredom in online English teaching in China, and findings indicated that most participants consider the online classes more boring than the face-to-face ones. This is also an important factor that influences instructor e-readiness level. Wang Yongliang [22] extracted four types of solutions from the data, including teacher-related solutions, task-related solutions, student-related solutions, and IT-related solutions. This study is helpful in dealing with practical online teaching problems. Based on the substitution augmentation modification redefinition model (SAMR) and the unified theory of acceptance and use of technology (UTAUT), Zou Cuiying, Li Ping and Jin Li [23] identified factors affecting students' satisfaction with and perceived learning performance in the practice of integrating smartphones in EFL classrooms. This study provided reference for teaching design with extensive use of smartphones in EFL classrooms. Yang Gao, Gang Zeng and WangYongliang [24] conducted the study on educational planning, teacher beliefs, and teacher practices during the

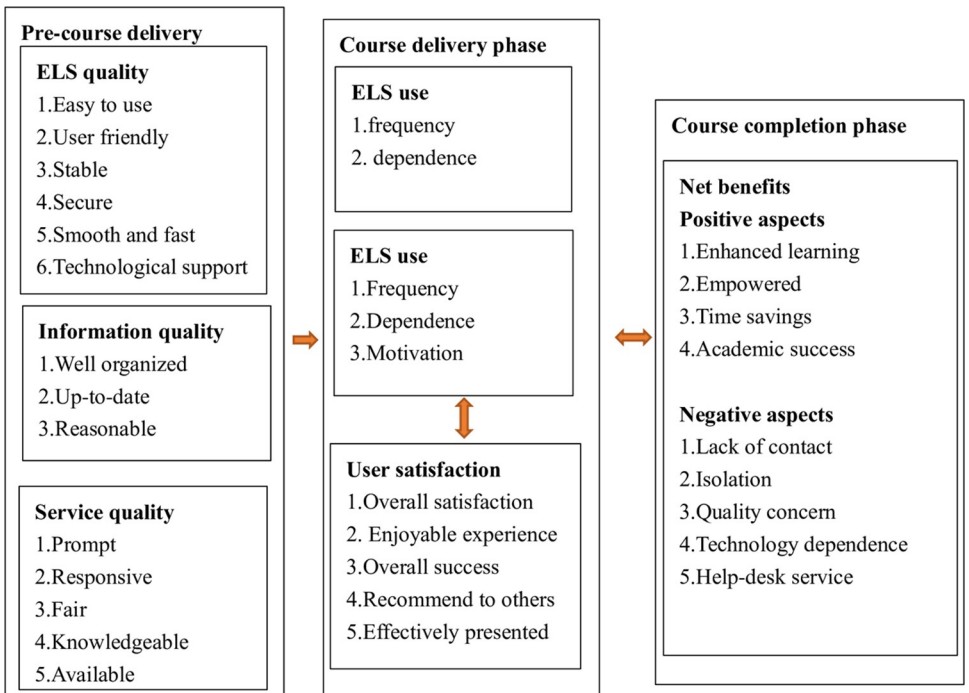

**Fig 2. Conceptual framework of e-readiness factors adapted from Glenda H. E. Gay (2016) for online student/ instructor e-readiness before, during and after course delivery.**

pandemic from a Chinese perspective, which provided insight on emergency remote teaching planning and implementation. All these studies provided guidance for the future research on figuring out the impediments, key learning elements and factors of online EFL classes.

## Student e-readiness for online EFL class

The notion of readiness for online learning among students has been proposed, defined and developed in some detail by Warner, Christie, and Choy [25]. The readiness for online language learning (OLL) is essential to determine learners' success and ability to achieve goals in an OLL course [26]. Learners with a higher level of readiness can be seen as having greater confidence and motivation or a desire to take responsibility or ownership of their learning [27].

Measurements for online learning readiness has been developed by scholars, such as Mcvay [28–30] from various dimension to validate online learning readiness scales, of which Glenda H. E. Gay [1] assessed from the perspectives of technical readiness, lifestyle readiness and pedagogical readiness, and others, including, but not limited to, measured from technology access, computer self-efficacy, self-directed learning, learner control, motivation for English learning and online communication self-efficacy.

Use of computer technology to learn a foreign language, though tempting, is determined by a number of personal and contextual factors [27]. Factors/variables that affects online learning readiness were validated [31] In 2003, two primary factors: comfort with e-learning and self-management of learning, which could predict online learning readiness success were identified [32]. Holsapple and Lee-Post [2] and Glenda H. E. Gay [1] deepened and developed the research on e-readiness level. Yu T., and Richardson J.C. [33] with exploratory factor analysis (EFA) and reliability analysis, listed twenty items from three competencies—social

competencies, communication competencies, and technical competencies to develop an effective instrument to test student e-readiness. Atqo Akmal et al. [34] measured student e-readiness from digital competence of users and digital infrastructure readiness. Dattibongs et al. [26] suggested four success factors which have relationship with the readiness for online language learning (OLL): attitude and motivation, self-regulated learning (SRL), English language self-efficacy, technology literacy and access.

Online language learning is different from online learning of other subjects [5]. However, currently, little attention has been paid to learner preparedness for online language learning [35]. In the online EFL class, language is the medium of instruction as well as the subject matter of online learning. Park, Moonyoung, and Jeong-Bae Son [36] explored a group of pre-service EFL teachers' information and communication technology experiences and their attitudes towards and perceived barriers to technology integration and found that use of digital technologies in the classroom has created a dynamic learning environment that could enhance learning and teaching. Rafiee Marzieh et al. [37] posits that current studies on readiness of OLL should be repeated with a great number of participants in different levels of English proficiency. Currently, there lacks researches conducted in inland provinces in China to assess student/instructor e-readiness for online English learning. Therefore, this exploratory study surveyed undergraduate students across 31 provinces in China to gauge student e-readiness for EFL class.

## Instructor e-readiness for online EFL class

Teacher readiness, or faculty/instructor readiness, is the willingness to prepare, effectively design and facilitate courses within an online environment, which is also the prerequisite for a student-satisfied online class. Cutri Ramona Maile et al. [38] pointed out that while some of the faculty felt forced to moved online, others felt well-prepared and enjoy it or come to enjoy it. However, in 2022, the United Nations Educational Scientific and Cultural Organization (UNESCO) reported that online education lacks quality because teachers are unprepared for online pedagogical knowledge [15].

The level of instructor e-readiness has been measured regarding to various dimensions. Martin F., Budhrani K., and Wang C. [39] measured by faculty attitudes about the importance of online teaching competencies and faculty's perceptions of their ability to confidently teach online. Gay [1] assessed with three scales before, during and after the course delivery: technical readiness, lifestyle readiness and pedagogical readiness to the e-learning system environment. Zou et al. [5] examined teacher readiness from six dimensions: technology access, computer self-efficacy, self-directed learning, learner control, motivation for English learning, and online communication self-efficacy, and found that teacher readiness was slightly low in Wuhan City. Investigation of Jwaifell M. et al. [40] revealed significant differences among teachers' readiness in a high degree.

Faculty members' willingness to teach online profits institutions of higher education [38]. However, these studies were mainly theoretical, focusing on explaining why certain skills are important, without measuring whether the language teachers were ready for online language teaching [5]. Thus, based on the previous study, this research paper is to examine instructor readiness level of online EFL class in inland China.

## Materials and methods

### Ethics statement

This research was investigation-oriented without revealing any specific personal information, so no ethical agreement was needed. Participants were surveyed online on a voluntary basis.

There is no actual examination on any of the participants themselves. The sentence: "The questionnaire is anonymous. The collected data will only be used in the future research. Any of the personal information will not be revealed. Thank you for your support." was listed in the front of the questionnaire. Therefore, no extra formal consent was obtained from the participants. Upon completion of the questionnaires, participants automatically granted use of their responses to the future research work.

## Data acquisition

This exploratory study was operated in inland China with two online questionnaires on the platform *WenJuanxing* (www.wjx.cn). It is a popular platform to collect questionnaire in China. Website of this research questionnaire is: *https://www.wjx.cn/vm/tUrrbPw.aspx#*. One questionnaire was for the students and the other one was for the instructors. Both of the questionnaires were composed of a demographic information form (see Tables 1 and 2), a e-readiness scale (see Tables 3 and 4) and some open-ended questions. The questionnaire was presented in a bilingual way: a Chinese sentence ahead and a corresponding English version followed.

This e-learning success model (ELSM) adopted and adapted from that of Holsapple and Lee-Post [2] and Gay [1]. The student e-readiness scale consists of 4 items for each of the three phases: pre-course, course delivery, and course completion phase (12 items in total), while the instructor e-readiness is composed of 5 items for the pre-course and course delivery phase respectively, and 6 items for the course completion phase (16 items in total). All items used a five-point Likert-type scale, ranging from "1 = strongly-disagree" to "5 = strongly-agree" (1 = strongly disagree, 2 = disagree, 3 = neutral, 4 = agree, 5 = strongly agree).

Students were undergraduates who were having online college classes, and instructors were teachers teaching online college English in the EFL context. 6024 students were surveyed in the student questionnaire. 110 of them had no English classes now, thus they submitted without answering more questions of the questionnaire. Therefore, there left 5914 valid questionnaires. Data were analyzed with SPSS. 85 Chinese universities/colleges were chosen as the research site. Participants surveyed were majoring in various majors (rounded 400), such as Education, Philosophy, Economics, Law, Food Safety, Literature, History, Sport, Architecture, Mechanism, Automobile, Agriculture, Medicine, Art, Military, Robotics Engineering, Human Resources Management, Accounting, Logistics Management, Auditing, International Business, Bioengineering, Translation and Interpretation, Economics, Sociology, Journalism and Communication, Marxism, Physics, Mathematics, etc.

1752 colleges English instructors were surveyed. They were teaching in different higher education institutes: directly under MOE, co-construction by provinces and ministries/under provincial department of education, municipal college, private undergraduate colleges, and state-owned vocational.

There were some open-ended questions, such as: "How will you be engaged in the online EFL class", "Please list some impediments of online EFL learning." and "Please list some recommendations on promoting online EFL class".

## Results

### RQ1: What's the student/instructor e-readiness level in online EFL class in inland China?

**Student E-readiness.**  Descriptive statistics from the SRS are presented in Table 3.

 

**Table 1. Demographics for student respondents.**

| Variable | Category | Frequency | Percentage (%) |
|---|---|---|---|
| **Gender** | Male | 3325 | 55.2 |
| | Female | 2699 | 44.8 |
| **Grade** | Freshman | 1453 | 24.12 |
| | Sophomore | 1691 | 28.07 |
| | Junior | 1567 | 26.01 |
| | Senior | 1313 | 21.8 |
| **Having English classes now** | Yes | **5914** | 98.17 |
| | **No** | **110** | 1.83 |
| **Regions to have online EFL class** | City | 5060 | 85.56 |
| | Countryside | 845 | 14.44 |
| **Province** | 31 | 5914 | 100 |
| **Property of the HEI** | Directly under MOE | 1407 | 23.79 |
| | Co-construction by provinces and ministries/Under provincial department of education | 3099 | 52.4 |
| | Municipal college | 437 | 7.39 |
| | Private undergraduate colleges | 684 | 11.57 |
| | State-owned vocational colleges | 287 | 4.85 |
| **Prior online EFL experiences (months)** | 1–6 | 1109 | 18.75 |
| | 7–12 | 1315 | 22.24 |
| | 13–24 | 903 | 15.27 |
| | 25+ | 394 | 6.66 |
| | 0 | 2193 | 37.08 |
| **Internet connectivity/access** | Mobilephone (3G/4G/5G) | 1304 | 22.05 |
| | Mobilephone(WIFI) | 1602 | 27.09 |
| | Computer(WIFI/Broadband) | 1224 | 20.7 |
| | Ipad(WIFI) | 1697 | 28.69 |
| | Others | 87 | 1.47 |
| **Willingness for further study** | Yes | 1355 | 22.91 |
| | No | 4559 | 77.09 |
| **Platform for online EFL class** | Tencent conference | 1243 | 21.02 |
| | QQ | 1193 | 20.17 |
| | Ding Talk (Ding Ding) | 1393 | 23.55 |
| | Chaoxing | 668 | 11.3 |
| | Rain Class | 846 | 14.31 |
| | Icve | 459 | 7.76 |
| | Others | 112 | 1.89 |
| **Total** | | **5914** | 100 |

The Cronbach's **α** cohorts during the whole teaching process was 0.809, which reveals that the questionnaire is of a high validity. The Cronbach's **α** cohorts were 0.865, 0.814 and 0.748 in pre-course phase, course delivery phase and course completion phase respectively.

The e-readiness level for online EFL class for the cohort of student respondents is 3.635. The Cronbach's **α** cohorts in the pre-course phase, course delivery course and course completion phase are 3.725, 3.666, and 3.513 respectively. They were lower than the ready level since a mean score of 4 on a five-point Likert-type scale could be seen as e-ready.

Devices using was the foundation to be e-ready. However, 14.44% students were from rural countryside in inland China, and a total of 22.05% of the surveyed students had online EFL classes with their mobile phones (3G/4G/5G), which to some extent, showed that they could

**Table 2. Demographics for teacher respondents.**

| Variable | Category | Frequency | Percentage |
|---|---|---|---|
| **Gender** | Female | 1162 | 66.32 |
| | Male | 590 | 33.68 |
| **Age** | 24-30y | 181 | 10.33 |
| | 31-35y | 348 | 19.86 |
| | 36-40y | 641 | 36.59 |
| | 41-45y | 445 | 25.4 |
| | 46y+ | 137 | 7.82 |
| **Property of the HEI** | Directly under MOE | 261 | 14.9 |
| | Co-construction by provinces and ministries/Under provincial department of education | 824 | 47.03 |
| | Municipal college | 185 | 10.56 |
| | Private undergraduate colleges | 365 | 20.83 |
| | State-owned vocational colleges | 117 | 6.68 |
| **Prior online EFL experiences (months)** | 1–6 | 227 | 12.96 |
| | 7–12 | 482 | 27.51 |
| | 13–24 | 679 | 38.76 |
| | 25+ | 362 | 20.66 |
| | 0 | 2 | 0.11 |
| **Mode of the online EFL class delivery** | Synchronous | 741 | 42.29 |
| | Asynchronous | 222 | 12.67 |
| | Combination of synchronous and asynchronous | 789 | 45.03 |
| **Platform for online EFL class** | Tencent conference | 389 | 22.2 |
| | QQ | 422 | 24.09 |
| | Ding Talk (Ding Ding) | 463 | 26.43 |
| | Chaoxing | 102 | 5.82 |
| | Rain Class | 230 | 13.13 |
| | Icve | 128 | 7.31 |
| | Others | 18 | 1.03 |
| Total | **1752** | | 100 |

not equip themselves with advanced devices or fluent internet connectivity such as fluent broadband, which, without doubt, led to a lower e-ready level. Students had to pay for the internet connectivity, which brought them financial burden.

11.7% of the students (strongly) disagreed with "*I think online EFL class arouses my interests in learning English*"; 16.53% students were (strongly) unsatisfied with the online EFL class, and 17.23% of them (strongly) disagreed with "*The online EFL class has gained an overall success*", which illustrated that the students who were having the EFL class didn't have a satisfied online EFL class from the perspectives either of self-experiences or teaching effects. In addition, of the validated answers in the questionnaire, 77.09% of them didn't have the willingness for further study, which led to low motivation to learn in the course delivery phase.

With low e-ready level in the prior two phases, e-ready level in the course completion phase was undoubtedly slightly low. The lowest score was 3.155 in "*I think that online EFL class has improved my English academic success*", and the second lowest from the bottom was 3.548 in "*I think that the online EFL class improved my English learning capabilities*", which showed that students were not satisfied with the online EFL classes.

Analysis of variance (see Fig 3) were applied to examine whether there were differences between genders, regions, universities, and prior experiences. The results showed that: no significant differences were revealed between different genders except for the following two items

Table 3. Means (M) and Cronbach's alpha(α) of the student e-readiness scale (N = 5914).

| Dimensions & items | M | α |
|---|---|---|
| Pre-course phase (PP) | 3.725 | 0.865 |
| PP1 I recognize that the device using for the online EFL class is easy to operate. | 3.862 | |
| PP2 I recognize that the device using for the online EFL class is user-friendly. | 3.776 | |
| PP3 I recognize that the device using for the online EFL class is stable and secure. | 3.726 | |
| PP4 I recognize that the device using for the online EFL class runs smoothly and fast. | 3.537 | |
| Course delivery phase (CDP) | 3.666 | 0.814 |
| CDP1 I recognize that we have high frequency of online EFL class, generally having 4 classes per week, each class lasting for 45 minutes. | 3.655 | |
| CDP2 I am satisfied with the online EFL class. | 3.711 | |
| CDP3 The online EFL class has gained an overall success. | 3.679 | |
| CDP4 I think online EFL class arouses my interests in learning English. | 3.620 | |
| Course completion phase (CCP) | 3.513 | 0.748 |
| CCP1 I think the online EFL class connects me with my instructors and classmates. | 3.624 | |
| CCP2 I concern for the teaching and learning quality of the online EFL class. | 3.576 | |
| CCP3 I think that online EFL class has improved my English academic success. | 3.155 | |
| CCP4 As for the problems encountered after the EFL class, I can deal with them without any prompt help-desk service. | 3.660 | |
| CCP5 I think that the online EFL class improved my English learning capabilities. | 3.548 | |
| The whole process | M = 3.635 | α = 0.809 |

All items were measured via a 5-point Likert scale: 1 = strongly disagree, 2 = disagree, 3 = neutral, 4 = agree, 5 = strongly agree

"*I concern for the teaching and learning quality of the online EFL class* (M±SD: Male: 2.46±1.21 Female: 2.38±1.20 F: 5.687 p: 0.017*)." and "*I think that online EFL class has improved my English academic success* (M±SD: Male: 2.81±1.37 Female: 2.89±1.39 F: 5.052 p: 0.025*)." Significant differences were revealed on students from different regions and universities, and students with or without prior experiences all revealed significance differences except for "*I recognize that the device using for the online EFL class runs smoothly and fast* (see Fig 3)."

**Instructor E-readiness.** The overall Cronbach's α cohorts during the whole teaching process was 0.839, which revealed that the questionnaire was of a high validity. The Cronbach's α cohorts were 0.868, 0.845 and 0.803 in pre-course phase, course delivery phase and course completion phase respectively.

The e-readiness level Cronbach's α cohort of instructor respondents was 3.708 out of a score of 5. Cronbach's α cohorts in the pre-course phase, course delivery course and course completion phase were 3.758, 3.786, and 3.581 respectively. They were slightly lower than the ready level since a mean score of 4 on a five-point Likert-type scale could be seen as e-ready, yet the overall e-readiness level was higher than that of the student. Student/instructor e-readiness level in the course completion phase was virtually the same: 3.513 for the students and 3.581 for the instructors.

Analysis of variance (see Fig 4) were applied to examine whether there were differences between genders, age, teaching methods, universities, and prior experiences, of which research results all showed significant differences.

## RQ2: What are the impediments for online EFL class?

One of many obstacles in predicting learner's online learning success was defining "success." Success for distance education can be viewed from multiple perspectives, each having its own

Table 4. Means (M) and Cronbach's alpha(α) of the instructor e-readiness scale (N = 1752).

| Dimensions & items | M | α |
|---|---|---|
| Pre-course phase (PP) | 3.758 | 0.868 |
| PP1 I recognize that the device using for the online EFL class is easy to operate. | 3.899 | |
| PP2 I recognize that the device using for the online EFL class is user-friendly. | 3.692 | |
| PP3 I recognize that the device using for the online EFL class is stable and secure. | 3.660 | |
| PP4 I recognize that the device using for the online EFL class runs smoothly and fast. | 3.781 | |
| Course delivery phase(CDP) | 3.786 | 0.845 |
| CDP1 I recognize that the online EFL course structure is reasonable. | 3.836 | |
| CDP2 I have effectively presented the online EFL course for my students. | 3.763 | |
| CDP3 I recognize that the online EFL course is up-to-date. | 3.761 | |
| CDP4 I can give prompt responses to students. | 3.834 | |
| CDP5 I feel enjoyable in the online EFL classroom. | 3.735 | |
| Course completion phase(CCP) | 3.581 | 0.803 |
| CCP1 I'm satisfied with the online EFL class. | 3.812 | |
| CCP2 I think that students in the online EFL classroom has been improved. | 3.581 | |
| CCP3 I think the online EFL class saves me a lot of time. | 3.725 | |
| CCP4 I think the online EFL class connects me with my students. | 3.191 | |
| CCP5 I concern for the overall teaching quality of the online EFL class. | 3.678 | |
| CCP6 I think that the online EFL class help s gain success in English teaching. | 3.584 | |
| CCP7 As for the problems encountered after the EFL class, I can deal with them without any prompt help-desk service. | 3.495 | |
| The whole process | M = 3.708 | α = 0.839 |

All items were measured via a 5-point Likert scale: 1 = strongly disagree, 2 = disagree, 3 = neutral, 4 = agree, 5 = strongly agree

definition and criteria [41]. Zou et al. [5] summarized challenges of online learning into six types,; Wahid Bakar Hamad [42] summarized challenges encountered by the teachers as: skills and training in online teaching, internet/ infrastructure, supporting resources, student engagement & feedback, good strategies in online remote, relevant pedagogy, and suitable online platform and that encountered by the students as: skills and training in online learning, internet/ infrastructure, supporting resources, and engagement & feedback. The questionnaire in this research collected attitudes towards the impediments for online EFL class from both students and instructors, which were divided into eight types. To present impediments for online EFL class more comprehensively, this research adapted that from Zou et al. [5] and covered more aspects according to the questionnaire (see Tables 5 and 6).

The unprecedented transition from traditional face-to-face class to online learning brought great challenges to both students and instructors. Impediments confronted by students and instructors were mainly from technical challenges, learning process, learning environments, self-control, health concern, learning materials, assignment, and learning effect and assessments.

Similar impediments in the pre-course phase, course delivery phase and course completion phase between students and instructors covered: expensive payment for the internet connectivity, weak and feeble signal, no or little interaction, absent-minded (student) in the class, no peers (student) to have face-to-face group work, lack of learning/teaching passion, poor eyesight/eye fatigue, no or little feedback from students/instructors, lack of instant help-desk service, and low learning efficiency, which is to some degree, consistent with the finding of Rafiq Karmila Rafiqah M. et al [15].

| Items | PP1 | PP2 | PP3 | PP4 | CDP1 | CDP2 | CDP3 | CDP4 | CCP1 | CCP2 | CCP3 | CCP4 | CCP5 |
|---|---|---|---|---|---|---|---|---|---|---|---|---|---|
| Gender(1=M/2=F) | | | | | | | | | | | | | |
| 1.0(n=3263) | 2.14±1.06 | 2.23±1.19 | 2.28±1.17 | 2.30±1.16 | 2.33±1.15 | 2.29±1.13 | 2.31±1.15 | 2.35±1.15 | 2.38±1.18 | 2.46±1.21 | 2.81±1.37 | 2.33±1.16 | 2.47±1.22 |
| 2.0(n=2651) | 2.14±1.10 | 2.22±1.20 | 2.27±1.19 | 2.29±1.18 | 2.36±1.19 | 2.29±1.18 | 2.33±1.19 | 2.39±1.18 | 2.37±1.19 | 2.38±1.20 | 2.89±1.39 | 2.35±1.18 | 2.43±1.21 |
| F | 0.025 | 0.065 | 0.019 | 0.105 | 1.206 | 0.047 | 0.681 | 1.216 | 0.372 | 5.687 | 5.052 | 0.571 | 1.319 |
| p | 0.873 | 0.799 | 0.889 | 0.746 | 0.272 | 0.828 | 0.409 | 0.27 | 0.542 | 0.017* | 0.025* | 0.45 | 0.251 |
| Region(1=City/2=Countryside) | | | | | | | | | | | | | |
| 1.0(n=5060) | 2.06±1.03 | 2.16±1.15 | 2.20±1.15 | 2.22±1.13 | 2.28±1.13 | 2.23±1.12 | 2.26±1.14 | 2.31±1.13 | 2.32±1.16 | 2.37±1.18 | 2.80±1.38 | 2.29±1.14 | 2.41±1.20 |
| 2.0(n=854) | 2.61±1.23 | 2.62±1.32 | 2.72±1.27 | 2.74±1.21 | 2.72±1.31 | 2.65±1.30 | 2.69±1.28 | 2.69±1.27 | 2.70±1.27 | 2.74±1.28 | 3.08±1.37 | 2.65±1.29 | 2.72±1.30 |
| F | 203.123 | 115.22 | 147.616 | 146.251 | 104.497 | 96.64 | 102.957 | 78.132 | 74.522 | 70.639 | 29.83 | 71.285 | 49.785 |
| p | 0.000** | 0.000** | 0.000** | 0.000** | 0.000** | 0.000** | 0.000** | 0.000** | 0.000** | 0.000** | 0.000** | 0.000** | 0.000** |
| University (1=Directly under MOE/2=Co-construction by provinces and ministries/Under provincial department of education/3=Municipal college/4=Private undergraduate colleges/5=State-owned | | | | | | | | | | | | | |
| 1.0(n=1407) | 2.28±1.15 | 2.31±1.24 | 2.36±1.20 | 2.38±1.20 | 2.46±1.19 | 2.44±1.19 | 2.48±1.22 | 2.50±1.21 | 2.54±1.22 | 2.56±1.23 | 3.04±1.37 | 2.48±1.20 | 2.55±1.23 |
| 2.0(n=3099) | 1.99±0.95 | 2.13±1.13 | 2.17±1.12 | 2.19±1.11 | 2.20±1.11 | 2.15±1.09 | 2.16±1.10 | 2.21±1.10 | 2.20±1.12 | 2.23±1.13 | 2.52±1.31 | 2.18±1.11 | 2.28±1.17 |
| 3.0(n=437) | 2.33±1.20 | 2.35±1.28 | 2.49±1.26 | 2.49±1.26 | 2.63±1.22 | 2.51±1.22 | 2.58±1.21 | 2.61±1.19 | 2.67±1.21 | 2.67±1.26 | 3.25±1.33 | 2.73±1.23 | 2.75±1.23 |
| 4.0(n=684) | 2.32±1.16 | 2.35±1.23 | 2.37±1.23 | 2.39±1.18 | 2.48±1.21 | 2.36±1.19 | 2.48±1.20 | 2.54±1.22 | 2.55±1.20 | 2.73±1.25 | 3.36±1.34 | 2.46±1.21 | 2.71±1.26 |
| 5.0(n=287) | 2.30±1.31 | 2.30±1.33 | 2.36±1.30 | 2.47±1.37 | 2.57±1.27 | 2.48±1.29 | 2.52±1.28 | 2.63±1.22 | 2.59±1.28 | 2.69±1.29 | 3.62±1.29 | 2.51±1.29 | 2.75±1.30 |
| F | 31.296 | 10.173 | 13.064 | 13.652 | 27.018 | 23.715 | 32.34 | 30.685 | 37.815 | 44.064 | 113.83 | 36.361 | 36.62 |
| p | 0.000** | 0.000** | 0.000** | 0.000** | 0.000** | 0.000** | 0.000** | 0.000** | 0.000** | 0.000** | 0.000** | 0.000** | 0.000** |
| Prior experiences (1=1-6ms/2=7-12ms/3=13-24ms/4=25ms+/5=0) | | | | | | | | | | | | | |
| 1.0(n=62) | 1.71±0.52 | 1.82±0.53 | 1.94±0.65 | 2.08±0.68 | 2.26±0.87 | 2.05±0.71 | 2.55±1.04 | 2.19±0.83 | 2.44±0.93 | 1.90±0.65 | 3.19±1.04 | 2.69±1.06 | 2.79±1.04 |
| 2.0(n=17) | 1.88±0.99 | 1.82±1.01 | 2.00±0.79 | 2.00±0.79 | 2.47±1.01 | 2.35±0.86 | 2.59±0.87 | 2.12±0.78 | 2.76±1.30 | 1.82±0.53 | 3.29±1.31 | 2.29±0.92 | 2.71±0.99 |
| 3.0(n=7) | 1.86±0.38 | 2.14±0.38 | 2.14±0.38 | 2.43±0.53 | 2.00±0.58 | 1.86±0.38 | 1.86±0.38 | 1.57±0.53 | 2.43±0.79 | 1.71±0.49 | 3.43±0.98 | 2.43±0.53 | 2.86±0.38 |
| 4.0(n=4) | 1.75±0.50 | 1.75±0.96 | 1.75±0.96 | 2.75±0.96 | 2.50±1.00 | 2.75±0.96 | 2.75±0.96 | 2.75±0.96 | 2.25±0.50 | 2.00±1.41 | 4.00±0.82 | 3.00±1.15 | 2.50±1.73 |
| 5.0(n=10) | 2.40±0.97 | 2.20±0.63 | 2.20±0.42 | 2.80±1.03 | 3.10±0.99 | 2.40±0.70 | 2.90±0.74 | 2.60±0.70 | 3.30±0.82 | 2.20±0.63 | 2.90±1.10 | 2.30±0.67 | 2.50±1.08 |
| F | 2.352 | 1.069 | 0.586 | 3.035 | 2.282 | 1.875 | 1.318 | 2.242 | 1.998 | 0.738 | 0.826 | 0.985 | 0.251 |
| p | 0.059 | 0.376 | 0.673 | 0.021* | 0.066 | 0.121 | 0.269 | 0.07 | 0.101 | 0.568 | 0.512 | 0.419 | 0.908 |

**Fig 3. Student e-readiness differences with analysis of variance.**

These impediments explained why the e-readiness level during pre-course phase, course delivery phase, and course completion phase were lower than the ready level, which was also the guidance for future teaching to solve or avoid such impediments during the online EFL class.

| Items | PP1 | PP2 | PP3 | PP4 | CDP1 | CDP2 | CDP3 | CDP4 | CDP5 | CCP1 | CCP2 | CCP3 | CCP4 | CCP5 | CCP6 | CCP7 |
|---|---|---|---|---|---|---|---|---|---|---|---|---|---|---|---|---|
| Gender(1=M/2=F) | | | | | | | | | | | | | | | | |
| 1.0(n=590) | 2.37±1.19 | 2.56±1.24 | 2.61±1.26 | 2.46±1.24 | 2.34±1.25 | 2.47±1.26 | 2.49±1.27 | 2.33±1.22 | 2.40±1.25 | 2.43±1.22 | 2.65±1.25 | 2.44±1.21 | 2.97±1.29 | 2.50±1.22 | 2.65±1.25 | 2.67±1.24 |
| 2.0(n=1162) | 1.96±0.98 | 2.18±1.10 | 2.20±1.10 | 2.10±1.10 | 2.07±1.11 | 2.12±1.10 | 2.11±1.08 | 2.08±1.08 | 2.18±1.13 | 2.30±1.15 | 2.19±1.13 | | 2.73±1.34 | 2.23±1.12 | 2.30±1.14 | 2.42±1.19 |
| F | 60.029 | 41.501 | 50.009 | 38.136 | 20.964 | 35.65 | 43.957 | 19.132 | 31.318 | 17.315 | 32.572 | 18.991 | 12.489 | 21.01 | 36.283 | 17.579 |
| p | 0.000** | 0.000** | 0.000** | 0.000** | 0.000** | 0.000** | 0.000** | 0.000** | 0.000** | 0.000** | 0.000** | 0.000** | 0.000** | 0.000** | 0.000** | 0.000** |
| Age(1=24-30y/2=31-35y/3=36-40y/4=41-45y/5=46y+) | | | | | | | | | | | | | | | | |
| 1.0(n=181) | 2.35±1.26 | 2.50±1.28 | 2.56±1.30 | 2.43±1.25 | 2.12±1.16 | 2.20±1.19 | 2.31±1.18 | 2.19±1.17 | 2.22±1.16 | 2.41±1.26 | 2.66±1.29 | 2.38±1.20 | 3.18±1.30 | 2.51±1.18 | 2.69±1.24 | 2.68±1.25 |
| 2.0(n=348) | 2.09±1.09 | 2.33±1.22 | 2.40±1.14 | 2.19±1.14 | 2.17±1.23 | 2.29±1.17 | 2.27±1.18 | 2.15±1.17 | 2.20±1.10 | 2.35±1.19 | 2.51±1.19 | 2.31±1.21 | 3.02±1.30 | 2.40±1.19 | 2.49±1.21 | 2.61±1.24 |
| 3.0(n=641) | 1.97±0.95 | 2.19±1.06 | 2.20±1.12 | 2.06±1.07 | 2.08±1.11 | 2.12±1.07 | 2.14±1.09 | 2.12±1.09 | 2.07±1.09 | 2.17±1.11 | 2.29±1.15 | 2.20±1.09 | 2.67±1.32 | 2.21±1.10 | 2.29±1.12 | 2.37±1.12 |
| 4.0(n=445) | 2.05±1.01 | 2.28±1.13 | 2.27±1.11 | 2.22±1.15 | 2.18±1.15 | 2.27±1.18 | 2.22±1.20 | 2.13±1.14 | 2.24±1.19 | 2.15±1.10 | 2.30±1.14 | 2.19±1.15 | 2.64±1.30 | 2.27±1.15 | 2.35±1.20 | 2.48±1.23 |
| 5.0(n=137) | 2.58±1.26 | 2.66±1.29 | 2.78±1.35 | 2.77±1.29 | 2.52±1.26 | 2.58±1.38 | 2.60±1.27 | 2.47±1.36 | 2.52±1.32 | 2.65±1.36 | 2.87±1.31 | 2.69±1.25 | 2.99±1.37 | 2.59±1.32 | 2.65±1.30 | 2.74±1.32 |
| F | 12.215 | 6.254 | 9.563 | 12.678 | 4.054 | 4.909 | 4.669 | 2.812 | 4.956 | 7.156 | 10.461 | 6.083 | 9.997 | 5.243 | 6.233 | 4.995 |
| p | 0.000** | 0.000** | 0.000** | 0.000** | 0.003** | 0.001** | 0.001** | 0.024* | 0.001** | 0.000** | 0.000** | 0.000** | 0.000** | 0.000** | 0.000** | 0.001** |
| Teaching method(1=Synchronous/2=Asynchronous/3=Combination of the two) | | | | | | | | | | | | | | | | |
| 1.0(n=741) | 2.07±1.10 | 2.27±1.17 | 2.34±1.15 | 2.17±1.16 | 2.13±1.17 | 2.22±1.15 | 2.21±1.16 | 2.10±1.14 | 2.20±1.15 | 2.26±1.20 | 2.48±1.16 | 2.30±1.17 | 3.04±1.31 | 2.39±1.16 | 2.50±1.16 | 2.57±1.22 |
| 2.0(n=222) | 2.57±1.23 | 2.78±1.27 | 2.70±1.29 | 2.61±1.26 | 2.47±1.25 | 2.55±1.37 | 2.63±1.29 | 2.44±1.30 | 2.50±1.25 | 2.57±1.24 | 2.71±1.32 | 2.51±1.25 | 3.09±1.36 | 2.51±1.28 | 2.73±1.28 | 2.86±1.29 |
| 3.0(n=789) | 2.00±0.95 | 2.21±1.09 | 2.24±1.14 | 2.15±1.10 | 2.11±1.12 | 2.17±1.12 | 2.15±1.11 | 2.15±1.12 | 2.09±1.10 | 2.19±1.10 | 2.28±1.17 | 2.19±1.12 | 2.51±1.28 | 2.21±1.12 | 2.25±1.17 | 2.34±1.15 |
| F | 26.288 | 21.926 | 13.815 | 14.97 | 8.744 | 9.251 | 15.154 | 7.678 | 11.603 | 9.229 | 12.775 | 7.255 | 38.301 | 8.198 | 17.705 | 18.063 |
| p | 0.000** | 0.000** | 0.000** | 0.000** | 0.000** | 0.000** | 0.000** | 0.000** | 0.000** | 0.000** | 0.000** | 0.001** | 0.000** | 0.000** | 0.000** | 0.000** |
| University (1=Directly under MOE/2=Co-construction by provinces and ministries/Under provincial department of education/3=Municipal college/4=Private undergraduate colleges/5=State-owned vocational colleges) | | | | | | | | | | | | | | | | |
| 1.0(n=261) | 1.91±1.29 | 1.95±1.29 | 2.02±1.28 | 1.95±1.31 | 2.08±1.18 | 2.30±1.29 | 2.27±1.21 | 2.17±1.23 | 2.19±1.20 | 2.24±1.25 | 2.46±1.27 | 2.33±1.29 | 3.48±1.33 | 2.39±1.30 | 2.54±1.19 | 2.49±1.20 |
| 2.0(n=824) | 1.90±0.86 | 2.16±1.01 | 2.17±1.01 | 2.03±0.98 | 2.01±1.07 | 2.08±1.05 | 2.08±1.07 | 2.00±1.03 | 2.04±1.04 | 2.11±1.06 | 2.26±1.09 | 2.11±1.03 | 2.75±1.30 | 2.19±1.05 | 2.27±1.09 | 2.38±1.14 |
| 3.0(n=185) | 2.55±1.22 | 2.76±1.31 | 2.80±1.28 | 2.56±1.25 | 2.65±1.26 | 2.47±1.21 | 2.57±1.30 | 2.55±1.24 | 2.74±1.25 | 2.74±1.33 | 2.60±1.26 | 2.52±1.22 | | 2.58±1.27 | 2.75±1.40 | 2.79±1.33 |
| 4.0(n=365) | 2.30±1.04 | 2.50±1.14 | 2.56±1.17 | 2.51±1.16 | 2.22±1.17 | 2.29±1.19 | 2.33±1.16 | 2.28±1.17 | 2.24±1.16 | 2.33±1.16 | 2.47±1.19 | 2.35±1.19 | 2.63±1.30 | 2.36±1.15 | 2.37±1.16 | 2.53±1.21 |
| 5.0(n=117) | 2.58±1.22 | 2.81±1.23 | 2.82±1.30 | 2.74±1.31 | 2.46±1.37 | 2.67±1.36 | 2.49±1.31 | 2.35±1.38 | 2.53±1.36 | 2.50±1.34 | 2.77±1.36 | 2.56±1.31 | 2.74±1.34 | 2.60±1.33 | 2.74±1.39 | 2.89±1.39 |
| F | 27.78 | 25.982 | 25.668 | 26.264 | 14.339 | 10 | 9.82 | 11.683 | 11.153 | 13.178 | 9.747 | 10.48 | 22.151 | 7.225 | 9.75 | 7.847 |
| p | 0.000** | 0.000** | 0.000** | 0.000** | 0.000** | 0.000** | 0.000** | 0.000** | 0.000** | 0.000** | 0.000** | 0.000** | 0.000** | 0.000** | 0.000** | 0.000** |
| Prior experiences (1=1-6ms/2=7-12ms/3=13-24ms/4=25ms+/5=0) | | | | | | | | | | | | | | | | |
| 1.0(n=227) | 2.47±1.18 | 2.63±1.24 | 2.64±1.32 | 2.53±1.26 | 2.32±1.30 | 2.51±1.33 | 2.54±1.30 | 2.39±1.28 | 2.42±1.34 | 2.57±1.31 | 2.72±1.31 | 2.57±1.32 | 3.16±1.25 | 2.57±1.21 | 2.70±1.24 | 2.76±1.22 |
| 2.0(n=482) | 1.97±0.94 | 2.21±1.06 | 2.21±1.04 | 2.05±1.05 | 2.05±1.05 | 2.10±1.09 | 2.15±1.13 | 2.07±1.13 | 2.08±1.09 | 2.19±1.14 | 2.34±1.19 | 2.18±1.13 | 2.79±1.35 | 2.22±1.15 | 2.32±1.13 | 2.41±1.13 |
| 3.0(n=679) | 1.89±0.93 | 2.11±1.06 | 2.14±1.04 | 2.04±1.04 | 2.08±1.12 | 2.13±1.11 | 2.09±1.06 | 2.05±1.06 | 2.06±1.03 | 2.15±1.09 | 2.30±1.12 | 2.19±1.07 | 2.80±1.33 | 2.22±1.08 | 2.29±1.13 | 2.42±1.18 |
| 4.0(n=362) | 2.45±1.25 | 2.61±1.32 | 2.71±1.34 | 2.59±1.30 | 2.39±1.27 | 2.46±1.23 | 2.46±1.25 | 2.37±1.23 | 2.44±1.25 | 2.40±1.21 | 2.55±1.22 | 2.39±1.23 | 2.64±1.31 | 2.49±1.25 | 2.61±1.30 | 2.65±1.33 |
| 5.0(n=2) | 1.00±0.00 | 2.00±1.41 | 2.00±1.41 | 1.50±0.71 | 1.50±0.71 | 1.50±0.71 | 1.50±0.71 | 1.50±0.71 | 1.50±0.71 | 1.00±0.00 | 2.00±1.41 | 2.00±1.41 | 2.00±1.41 | 1.50±0.71 | 1.50±0.71 | 1.50±0.71 |
| F | 26.732 | 17.113 | 20.226 | 21.661 | 6.842 | 9.881 | 11.117 | 7.814 | 10.312 | 7.603 | 7.505 | 6.585 | 5.865 | 6.648 | 8.782 | 5.978 |
| p | 0.000** | 0.000** | 0.000** | 0.000** | 0.000** | 0.000** | 0.000** | 0.000** | 0.000** | 0.000** | 0.000** | 0.000** | 0.000** | 0.000** | 0.000** | 0.000** |

**Fig 4. Instructor e-readiness differences with analysis of variance.**

**Table 5. Categories of challenges reported by students.**

| Items | Technical challenges | Learning process | Learning environment | Self-control | Health concern | Learning Materials | Assignment | Effect and assessment |
|---|---|---|---|---|---|---|---|---|
| 1 | payment | interaction | solitary | motivation | poor eyesight | various materials | less group work | test |
| 2 | internet connectivity | pronunciation | no peers | self-disciplinary | sit for too long | boring | targeted | help-desk service |
| 3 | techno- skills | listening/speaking level | no inspectors | interests | no exercise | too easy/ difficult | online homework | practice |
| 4 | apps/software | vocabulary/grammar | teaching styles | confidence | | out of date | feedback from teachers | response |
| 5 | weak and feeble signal | long &difficult sentences | teacher presence | Learning passion | | | | low efficiency |
| 6 | | absent-minded | noisy | | | | | examination |

## RQ3: What are the learning elements for promoting e-learning success in online EFL class?

To study students'/instructors' attitudes towards online EFL learning elements, a multiple-choice question designed according to different phases of online EFL class was conducted. They should choose 6 out of the 15 items as the six most valuable online EFL learning elements (see Tables 7 and 8).

Students thought that in the pre-course phase, "*being self-discipline and properly managing time* (response rate: 7.37%; universalization rate: 53.04%)" is the most important, which illustrated that students had been aware of the importance of self-disciplinary studies for online learning. As for the course delivery phase, students regarded "*participating in student-student communications* (response rate: 7.39%; universalization rate: 53.18%)" "t*eacher presence* (response rate: 7.72%; universalization rate: 55.55%)" and "t*eacher-student interaction* (response rate: 8.27%; universalization rate: 59.5%)" as the most valuable ones. This showed that students wanted to interact with instructors and involved in the class and that they wanted to see the facial emotions or gestures from the instructors. In the course delivery phase, students wanted to "*review course content* (response rate: 7.71%; universalization rate: 55.5%)" and "*practice problem-solving ability* (response rate: 7.59%; universalization rate: 54.62%)". This showed that students want to apply what they have learned in the class into practice and foster the ability to solve practical problems.

Instructors, in the pre-course phase, agreed that "*fluent and free internet connect* (response rate: 7.66%; universalization rate: 43.04%)" was the most valuable online EFL learning

**Table 6. Categories of challenges reported by instructors.**

| Items | Technical challenges | Teaching process | Learning environment | Professional ability | Health concern | Learning Materials | Assignment | Effect and assessment |
|---|---|---|---|---|---|---|---|---|
| 1 | payment | group activities | no peers (student) | improve | no exercise | burden | check/examine | no proper assessing methods |
| 2 | techno- skills | student participation | communication with students/colleges | train/ cultivate | eye fatigue | common topics | convenient | uncertain of students' learning effect |
| 3 | internet lag | inspection | inspection | teaching experiences | tired | content design | student feedback | examination |
| 4 | weak and feeble signal | hang up/absent-minded | out of control | teaching passion | lack of enough rest | time-consuming | copy the answers for the internet | help-desk service |
| 5 | PPT design with new devices | interaction | | | | | terrible | |

**Table 7. Six most valuable online EFL learning elements from the students' view (15 items in total).**

|  | Items | Responsivity | | Universalization rate (%; n = 5914) |
|---|---|---|---|---|
|  |  | n | Response rate (%) |  |
| 1 | H. Teacher-student interaction | 3519 | 8.27 | 59.5 |
| 2 | G. Participating in student-student communications | 3285 | 7.72 | 55.55 |
| 3 | I. Reviewing course content | 3282 | 7.71 | 55.5 |
| 4 | N. Practicing problem-solving ability | 3230 | 7.59 | 54.62 |
| 5 | E. Teacher presence | 3145 | 7.39 | 53.18 |
| 6 | C. Being self-discipline and properly managing time | 3137 | 7.37 | 53.04 |
| 7 | K. Obtaining feedback from professor | 3119 | 8.27 | 52.72 |
| 8 | D. A silent place for studying | 3118 | 7.32 | 52.72 |
| 9 | A. Understanding the course materials | 3065 | 7.20 | 51.83 |
| 10 | B. Fluent and free internet connect | 2934 | 6.89 | 49.61 |
| 11 | J. Completing assignments | 2477 | 5.82 | 41.88 |
| 12 | F. Understanding the professor | 2455 | 5.77 | 41.51 |
| 13 | M. Taking practice examinations | 2409 | 5.66 | 40.73 |
| 14 | L. Obtaining feedback from assessed work | 2388 | 5.61 | 40.38 |
| 15 | O. Others | 1006 | 2.36 | 17.01 |
| Total | | 42569 | 100 | 719.80 |

**Note:** Pre-course phase: A B C D

Course delivery phase: E F G H

Course completion phase: I J K L M N O

**Table 8. Six most valuable online EFL learning elements from the instructors' view (15 items in total).**

|  | Items | Responsivity | | Universalization rate (%; n = 1752) |
|---|---|---|---|---|
|  |  | n | Response rate (%) |  |
| 1 | L. Obtaining feedback from assessed work | 1074 | 10.92 | 61.3 |
| 2 | M. Taking practice examinations | 1059 | 10.76 | 60.45 |
| 3 | H. Teacher-student interaction | 873 | 8.87 | 49.83 |
| 4 | N. Practicing problem-solving ability | 797 | 8.10 | 45.49 |
| 5 | B. Fluent and free internet connect | 754 | 7.66 | 43.04 |
| 6 | E. Teacher presence | 744 | 7.56 | 42.47 |
| 7 | C. Being self-discipline and properly managing time | 704 | 7.16 | 40.18 |
| 8 | K. Obtaining feedback from professor | 663 | 6.74 | 37.84 |
| 9 | G. Participating in student-student communications | 647 | 6.58 | 36.93 |
| 10 | F. Understanding the professor | 529 | 5.38 | 30.19 |
| 11 | A. Understanding the course materials | 525 | 5.24 | 29.97 |
| 12 | J. Completing assignments | 508 | 5.16 | 29 |
| 13 | I. Reviewing course content | 477 | 4.85 | 27.23 |
| 14 | D. A silent place for studying | 453 | 4.60 | 25.86 |
| 15 | O. Others | 32 | 0.33 | 1.83 |
| Total | | 9839 | 100 | 561.59 |

**Note:** Pre-course phase: A B C D

Course delivery phase: E F G H

Course completion phase: I J K L M N O

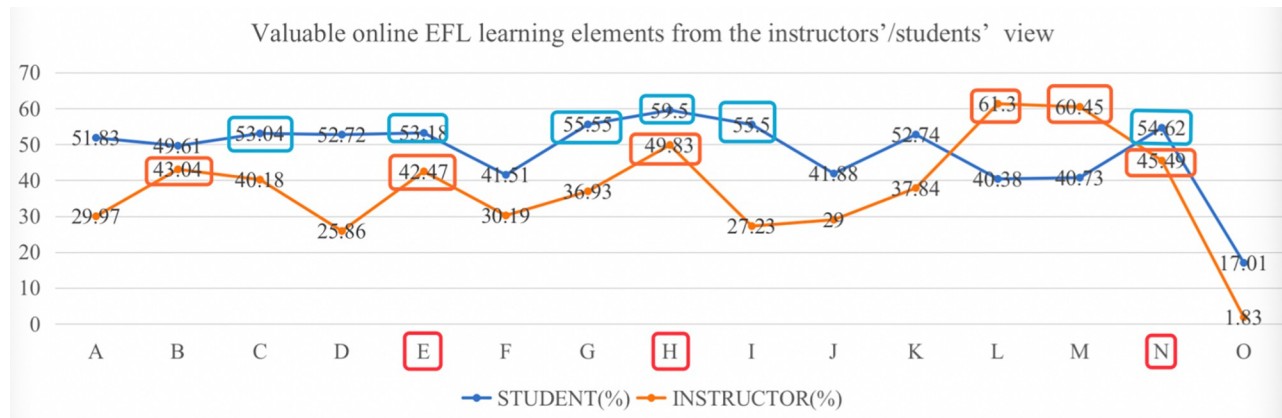

**Fig 5. Valuable online EFL learning elements from the instructors'/students' view.**

elements while in the course delivery phase, "*teacher-student interaction* (response rate: 8.87%; universalization rate: 49.83%)" and "*teacher presence* (response rate: 7.56%; universalization rate: 42.47%)" were the most valuable elements. In course completion phase, "*obtaining feedback from assessed work* (response rate: 10.92%; universalization rate: 61.3%)" "*taking practice examinations* (response rate: 10.76%; universalization rate: 60.45%)" and "*practicing problem-solving ability* (response rate: 8.10%; universalization rate: 45.49%)" were the most valuable learning elements.

Students and instructors had the same attitude towards three elements: "*teacher-student interaction*", "*practicing problem-solving ability*" and "*teacher presence*" (see Fig 5), which showed that students and instructors want to involve in the class through interaction and improve the problem-solving ability after the EFL class.

## RQ 4: What are the recommendations for promoting e-learning success in online EFL class?

Based on the prior data analysis of the impediments encountered by the students and instructors, this research summarized recommendations from the students and instructors to promote e-learning success in online EFL class, which were mainly categorized into seven types: (1) students: infrastructure and technology, learning process, content, curriculum design, teacher skills, service and assessment (2) instructors: infrastructure and technology, human resources, teaching quality, content and services, curriculum design, teacher skills, and assessment (see Tables 9 and 10).

As for the students, they needed to pay for the online learning platforms and internet connectivity, so some of them suggested that schools provide free infrastructure for them so as to release their financial burden. During the learning process, students pointed out that with student-instructor interaction and high learning interest, being self-discipline and concentrated also mattered. Meanwhile, teacher presence and instructors' personal charm were other key elements. For those humorous instructors, students in their online EFL class would have higher interests. Students concerned about teaching quality and pointed out that instructors should offer rich learning materials/ contents to improve their reading/ listening skills and abilities, that instructors should list clearly the key and difficult points and teaching objectives in online EFL class, and that online EFL class should be integrated with moral education as well, with advanced teaching skills. As for the offline help-desk services and assessments, students advised that instructors give timely response and feedback for the assignments and the

**Table 9. Categories of recommendations reported by students.**

| Items | Infrastructure and technology | Learning process | Content | Curriculum design | Teacher skills | Services | Assessment |
|---|---|---|---|---|---|---|---|
| 1 | free online learning platforms | self-discipline | teaching quality | form of design | task-based teaching method | online/offline help-desk service | feedback |
| 2 | applications (apps) | concentrated | rich contents | clear objectives | professional ability | care | multi-dimension |
| 3 | free internet connectivity | interaction | reading/listening skills | innovative | natural | enough rest | assignment |
| 4 | free online learning devices | students' interest | key and difficult points | English movie/novel | assessment | instructor-student relation-ship | examination |
| 5 | technological support | teacher presence/charm | culture/moral education | | | encourage | |

assessments should be multi-dimensional. They also expressed that there should be less assignments so that they would have more time to rest.

As for instructors, they hoped that schools will give financial supports to equip free internet connectivity with free devices such as computers, ipads, and laptops, and if possible, cooperate with other universities to share resources. Unlike traditional face-to-face classes, online teaching brought convenience for instructors. Therefore, to improve teaching quality and perfect teaching content and service, instructors recommended online coordinated notes (students), online curriculum design, up-to-date teaching materials and instant help-desk services with advanced/popular teaching approaches. Meanwhile, instructors should also care about students' psychological health. Assessments for students should be comprehensive through assignments, quizzes, examination and feedback.

## Discussion

This study aimed at examining student/instructor e-readiness, online English teaching and learning impediments, key online EFL learning elements, and recommendations for promoting online EFL class.

Both the students' and instructors' e-readiness levels were slightly lower than the ready level: 3.635 for students and 3.708 for instructors. As for the students, with the approach of analysis of variance, data indicate that there is virtually no significant difference between genders except for the following two items: "I am concerned about the teaching and learning quality of the online EFL class (M±SD: Male: 2.46±1.21 Female: 2.38±1.20 F: 5.687 p: 0.017*)" and "I think that the online EFL class has improved my English academic success (M±SD: Male:

**Table 10. Categories of recommendations reported by instructors.**

| Items | Infrastructure and technology | Human resources | Teaching quality | Content and service | Curriculum design | Teacher skills | Assessment |
|---|---|---|---|---|---|---|---|
| 1 | internet connectivity | school leaders | online coordinated notes (students) | update teaching materials | online group activity | advanced/popular teaching approaches | assignment |
| 2 | computers/ipads | teachers | online curriculum design | help-desk service | objectives | training | summary |
| 3 | financial supports | students &peers | ppt | patient | online materials | | feedback |
| 4 | cooperation among universities | | teaching ideology | psychological health | online design with collogues | | quiz/examination |
| 5 | | | online free MOOCs | magazines and English newspaper | key and difficult points | | comprehensive |

2.81±1.37 Female: 2.89±1.39 F: 5.052 p: 0.025*).” Students with and without prior experience all revealed significant differences in all items except for: “I recognize that the device used for the online EFL class runs smoothly and fast.” There were no significant differences between regions and universities; however, for instructors, there were significant differences between genders, age, teaching methods, universities, and prior experiences.

However, there are impediments to online EFL learning. Unlike face-to-face instruction, online learning and teaching encountered unprecedented challenges from the following aspects: technical challenges, learning/teaching process, learning environment, self-control/ professional ability, health concerns, learning materials, assignments, and effect and assessment. The issues related to high payments, weak and feeble signals, less interaction, poor eyesight, low learning efficiency, less or no student/teacher feedback, and lack of help-desk services, need to be solved so that students and instructors can be better involved in the online EFL course.

Students and instructors posit that teacher presence, teacher-student interaction, and practicing problem-solving are valuable online EFL learning elements. Students also regard student-student interaction, course reviewing, and problem-solving ability as important elements to improve online studies in the EFL context. Based on this finding, instructors are expected to involve students by interacting with them more frequently and connecting theoretical learning with practice in daily life. In addition, instructors should review courses in time.

Based on prior data analysis of valuable online EFL learning elements and impediments for online EFL learning, students and instructors offer recommendations for promoting learning success in online EFL classes. Recommendations were collected and summarized so that administrators and teachers could refer to them in future teaching planning and implementation. Students and instructors recommend that free online learning platforms, high teaching quality, clear curriculum design, instant help-desk services, and effective feedback should be taken into consideration in future teaching design and implementation.

In contrast to prior studies, this study designed a questionnaire to collect data from 85 Chinese universities. Questionnaire were collected in more than 30 provinces and the sample size were virtually equally distribute in each province. Furthermore, this study followed the recommendations of Gay and Zou, who pointed out that items like “desk-help service” [1] and “prior online learning experiences” [5] should be covered in the future research. Moreover, the questionnaire covered more participants, making the research data more convincing.

This study has some limitations that should be addressed. Future studies should cover more of the online learning elements, and categories of the impediments should be divided more specifically to have a more comprehensive and deeper understanding of the research question. The sample size of students and instructors should be more appropriate in future studies. There is also a need to conduct research with an action research approach in the final online EFL class to examine whether the recommendations by students and instructors are effective.

## Supporting information

**S1 File.**
(ZIP)

## Acknowledgments

We would like to thank all instructors and students surveyed in this research. Special thanks were given to Dr. Xie Weiwei and Dr. Ma Qian, who gave us so much help during the revision process.

 

## Author Contributions

**Conceptualization:** Chunye Yang.

**Data curation:** Defeng Xu.

**Investigation:** Chunye Yang, Defeng Xu.

**Methodology:** Chunye Yang.

**Project administration:** Chunye Yang.

**Writing – original draft:** Chunye Yang.

**Writing – review & editing:** Defeng Xu.

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
