## [Decision Letter · Decision Letter 0]

2 Feb 2023

PONE-D-23-00431Predicting student and instructor e-readiness and promoting e-learning success in online EFL class during the COVID-19 pandemic: A case from ChinaPLOS ONE

Dear Dr. Yang,

Thank you for submitting your manuscript to PLOS ONE. After careful consideration, we feel that it has merit but does not fully meet PLOS ONE’s publication criteria as it currently stands. Therefore, we invite you to submit a revised version of the manuscript that addresses the points raised during the review process.

We look forward to receiving your revised manuscript.

Kind regards,

Ehsan Namaziandost

Academic Editor

PLOS ONE

Journal Requirements:

2. Thank you for including your ethics statement:  "This research was an investigation-oriented without revealing any specific personal information, so no ethical agreement was needed. Participants were recruited on a voluntary basis, and there was a sentence: “The questionnaire is anonymous, whose data will only be used in the future research, and not any personal information will be revealed. Thank you for your support.” Therefore, no extra formal consent was obtained from the participants. Upon completion of the questionnaires, they automatically granted use of their responses.".   

a. For studies reporting research involving human participants, PLOS ONE requires authors to confirm that this specific study was reviewed and approved by an institutional review board (ethics committee) before the study began. Please provide the specific name of the ethics committee/IRB that approved your study, or explain why you did not seek approval in this case.

b. Please provide additional details regarding participant consent. In the ethics statement in the Methods and online submission information, please ensure that you have specified (1) whether consent was informed and (2) what type you obtained (for instance, written or verbal, and if verbal, how it was documented and witnessed). If your study included minors, state whether you obtained consent from parents or guardians. If the need for consent was waived by the ethics committee, please include this information.

Reviewers' comments:

Reviewer's Responses to Questions

**Comments to the Author**

1. Is the manuscript technically sound, and do the data support the conclusions?

Reviewer #1: Yes

Reviewer #2: Partly

Reviewer #3: Yes

2. Has the statistical analysis been performed appropriately and rigorously? 

Reviewer #1: Yes

Reviewer #2: Yes

Reviewer #3: Yes

3. Have the authors made all data underlying the findings in their manuscript fully available?

Reviewer #1: Yes

Reviewer #2: No

Reviewer #3: Yes

4. Is the manuscript presented in an intelligible fashion and written in standard English?

Reviewer #1: No

Reviewer #2: No

Reviewer #3: Yes

5. Review Comments to the Author

Reviewer #1: Dear authors,

PONE-D-23-00431

This project is interesting. Ideas are clear and the writing is concise and argumentative. Please see my comments below for your reference.

1 Please briefly describe the research problem and the importance of the work in the Abstract.

2 The Introduction starts with a general overview of the topic, presents a rationale for the paper, establishes the research problem, and specifies the objectives. When it comes to Chinese context in respect of the negative impacts online education brings, this reference is recommended to underpin your viewpoint (Wang, Y. (2023). Probing into the boredom of online instruction among Chinese English Language teachers during the Covid-19 pandemic. Current Psychology.43(1):1-15.https://doi.org/10.1007/s12144-022-04223-3.)

3 The Literature Review provides a theoretical context for the research. It is suggested that this section needs to narrow its scope to the Chinese context by involving more burgeoning literature, such as a) Review of the book investigating dynamic relationships among individual difference variables in learning English as a foreign language in a virtual world, by M. Kruk. System 100:102531. doi: 10.1016/j.system.2021.102531? ; b) Gao, Y., Zeng, G. Wang, Y., Klan, A. & Wang, X. (2022). Exploring educational planning, teacher beliefs, and teacher practices during the pandemic: A study of science and technology-based universities in China. Front. in Psycho., 13:903244. doi: 10.3389/fpsyg.2022.903244).

4 Please carefully check the research gap presented in the Introduction and the Literature review, and then adjust them to make the research problem consistent throughout the whole study.

5 The Methodology is clear, but the research site and the samples need to be elaborated upon.

6 The Findings and Discussion summarize the results in clear and concise language. Please place your study within the context of previous studies more closely.

7 This paper features too many lengthy sentences. They not only make the paper unreadable and incomprehensible, but also lead to grammatical mistakes. Please do make every single sentence crystal clear and grammatically accurate first.

Reviewer #2: Given the number of instructors included in the study, there were arguably few students included in comparison, I wonder why that was. I mean, there were too few students compared to the number of instructors.

In page 9, the authors write: "In this adapted framework, the capitalized letters “I” and “S” refer to “instructor” and “student”". But I do not see these abbreviations being used in the literature review, perhaps these two paragraphs would be better located in the methodology? In any case, it is not evident why these two paragraphs are there. I think that this notation is not even used (or at least consistently used) when providing the results, conclusion or in the tables.

In page 11, it says "A total of 6024 students were withdrawn in the student questionnaire" withdrawn? Did you mean surveyed? Check the writing in this section, it is confusing.

In page 12, what does "Subjective questions" mean?

The authors did not include enough information about the data collection instrument in the methods section. Also, the research design, specifically the pre-course and course delivery phases were not described.

Formal results and discussion sections were not provided, the results are provided just below methodology without another main heading. I think that results may be presented in better manner and properly discussed.

Writing style should be improved, the manuscript is readable, but in some sections, it becomes difficult to understand what the authors are referring to. There are also minor grammar issues such as use of verb tenses, plural/singular, and use of articles.

The authors have declared that data are fully available but then they stated that "All relevant data are within the manuscript and its Supporting Information files", I believe that given PLOS best practices, data should be fully provided in a data repository. Tables are valuable but they are not enough to suggest that all data were made available.

Reviewer #3: Thank you for submitting your research paper for PLOS. This paper covers an important topic. While it is a well-written paper, there are some issues that need to be addressed.

1. The novelty of this paper is not clear as many research papers have been published in this area. Thus, the novelty of this paper should be further justified by highlighting main contributions to the existing literature in order to be published in a reputable journals that seek new knowledge and insights.

2. The authors need to add a paragraph at the end of the introduction to show the structure of the paper.

3. The authors are required to provide a description of the data analysis methodology at the beginning of the data analysis section.

4. The authors need to define the population of the study and sampling technique adopted. Also please justify why your chosen sampling technique and sample size are appropriate.

5. Theoretical and practical implications should be reported after the discussion section. It is recommended to rely on the findings to provide thorough implications.

6. The authors are advised to strengthen the literature review and discussion sections by including more recent and relevant literature. This includes but not limited to:

- Novel extension of the UTAUT model to understand continued usage intention of learning management systems: the role of learning tradition. Doi: https://doi.org/10.1007/s10639-021-10758-y

- Towards a Sustainable Adoption of E-Learning Systems: The Role of Self-Directed Learning. Doi: https://doi.org/10.28945/4980

- Developing a Holistic Success Model for Sustainable E-Learning: A Structural Equation Modeling Approach. Doi: https://doi.org/10.3390/su13169453

- Evaluating E-learning systems success: An empirical study. Doi: https://doi.org/10.1016/j.chb.2019.08.004

6. PLOS authors have the option to publish the peer review history of their article (what does this mean?). If published, this will include your full peer review and any attached files.

Reviewer #1: **Yes: **Yongliang Wang

Reviewer #2: **Yes: **Juan D. Machin-Mastromatteo

Reviewer #3: **Yes: **Ahmad Samed Al-Adwan

---

## [Author Response · Author response to Decision Letter 0]

1 Mar 2023

PONE-D-23-00431

Predicting student and instructor e-readiness and promoting e-learning success in online EFL class during the COVID-19 pandemic: A case from China

Response to editors and reviewers

Dear Editors and Reviews ,

We feel great thanks for your professional review work on our article. We appreciate the reviewers for providing us the opportunity for some clarifications and modifications during this reviewing process. According to your nice suggestions, we have made extensive corrections to our previous manuscript, the detailed corrections are listed below.

We are looking forward to your reply.

Kind regards,

Chunye Yang

Journal Requirements:

Response: Thank you for your kind suggestion. We refer to the templates and make some changes in the revised manuscript to meet the requirements.

2.Thank you for including your ethics statement:  "This research was an investigation-oriented without revealing any specific personal information, so no ethical agreement was needed. Participants were recruited on a voluntary basis, and there was a sentence: “The questionnaire is anonymous, whose data will only be used in the future research, and not any personal information will be revealed. Thank you for your support.” Therefore, no extra formal consent was obtained from the participants. Upon completion of the questionnaires, they automatically granted use of their responses.".   

a. For studies reporting research involving human participants, PLOS ONE requires authors to confirm that this specific study was reviewed and approved by an institutional review board (ethics committee) before the study began. Please provide the specific name of the ethics committee/IRB that approved your study, or explain why you did not seek approval in this case.

Response: Thank you for the comments. We are sorry that the research sought no approval of any institutional review board (ethics committee), because a. participants surveyed of the questionnaire in this research were all on a voluntary basis; b. the questionnaire is anonymous; any of the personal information will not be revealed in the future, and data collected from the questionnaire were only used in the future research; c. most importantly, there is not any actual examination on any of the participants themselves. Data were collected through an online questionnaire, the website is https://www.wjx.cn/vm/tUrrbPw.aspx#. 

Response: Thank you for the suggestion. We have amended the ethnic statements in a clearer way and add it to “Ethics Statement” field of the submission form (via “Edit Submission”, See Line232-239).

The revised ethics statement is:

This research was investigation-oriented without revealing any specific personal information, so no ethical agreement was needed. Participants were surveyed online on a voluntary basis. There is no actual examination on any of the participants themselves. The sentence: “The questionnaire is anonymous. The collected data will only be used in the future research. Any of the personal information will not be revealed. Thank you for your support.” was listed in the front of the questionnaire. Therefore, no extra formal consent was obtained from the participants. Upon completion of the questionnaires, participants automatically granted use of their responses to the future research work.

b. Please provide additional details regarding participant consent. In the ethics statement in the Methods and online submission information, please ensure that you have specified (1) whether consent was informed and (2) what type you obtained (for instance, written or verbal, and if verbal, how it was documented and witnessed). If your study included minors, state whether you obtained consent from parents or guardians. If the need for consent was waived by the ethics committee, please include this information.

Response: We earnestly appreciate the valuable suggestion. Ethics statement of the questionnaire is written in the front of the questionnaire in Chinese (see Fig. 1), any one of the participants was acquiescently regarded as “known/agreed”. 

There were no minors in the survey. Participants were mainly undergraduates, who were in general, about 18 years old, and university instructors.

Fig. 1 Ethics statement of the questionnaire in Chinese in the front of the questionnaire

Response: Thank you for the comments. The research reported no medical records.

3.We suggest you thoroughly copyedit your manuscript for language usage, spelling, and grammar. If you do not know anyone who can help you do this, you may wish to consider employing a professional scientific editing service.  

Response: We earnestly appreciate the valuable suggestion. We have tried our best to re-read and make some changes to the manuscript to improve the manuscript language. We shortened/removed some of the lengthy sentences to make them clearer. These changes will not influence the content and framework of the article. Here, we did not list the changes. Again, we appreciate for your warm work earnestly.

Moreover，we sought help from Editage. The editing certificate is as follows:

Reviewer #1: Dear authors,

PONE-D-23-00431

This project is interesting. Ideas are clear and the writing is concise and argumentative. Please see my comments below for your reference.

1 Please briefly describe the research problem and the importance of the work in the Abstract.

Response: We earnestly appreciate the valuable suggestion. We have revised this part according to the Reviewer’s suggestion (See Line 12-36).

The revised version is as follows:

(a)research problem：This research, based on E-learning Success Model (ELSM), seeks to exam the e-readiness level of students/instructors, judge the impediments that students/instructors encountered in different phases—pre-course delivery, course delivery, and course completion phase of the online EFL class, search for valuable online learning elements, and recommend recommendations for promoting e-learning success in online-EFL class. 

(b)importance of the study: The outcomes of this research have theoretical and practical implications for researchers and higher education institutions (HEIs). And during an unprecedented time, administrators and instructors will have the insights on implementing emergency remote teaching.

2 The Introduction starts with a general overview of the topic, presents a rationale for the paper, establishes the research problem, and specifies the objectives. When it comes to Chinese context in respect of the negative impacts online education brings, this reference is recommended to underpin your viewpoint (Wang, Y. (2023). Probing into the boredom of online instruction among Chinese English Language teachers during the Covid-19 pandemic. Current Psychology.43(1):1-15.https://doi.org/10.1007/s12144-022-04223-3.)

Response: We sincerely appreciate the comments. The recommended reference are valuable and helpful for the research and we have added these literature in the literature review while revising the manuscript according to the suggestion (see Line 144-155).

3 The Literature Review provides a theoretical context for the research. It is suggested that this section needs to narrow its scope to the Chinese context by involving more burgeoning literature, such as a) Review of the book investigating dynamic relationships among individual difference variables in learning English as a foreign language in a virtual world, by M. Kruk. System 100:102531. doi: 10.1016/j.system.2021.102531? ; b) Gao, Y., Zeng, G. Wang, Y., Klan, A. & Wang, X. (2022). Exploring educational planning, teacher beliefs, and teacher practices during the pandemic: A study of science and technology-based universities in China. Front. in Psycho., 13:903244. doi: 10.3389/fpsyg.2022.903244).

Response: We sincerely appreciate the valuable comments. We have checked the literature carefully and added more references. These include but not limited to the literature you have suggested above in the revised manuscript (See 144-163).

4 Please carefully check the research gap presented in the Introduction and the Literature review, and then adjust them to make the research problem consistent throughout the whole study.

Response: Thanks for your suggestion. We tried our best to make the research problem consistent throughout the whole study in the revised manuscript. In the introduction, we list the research questions and research structure of the article. In the literature review, we add more literature to make the introduction closely related to the literature review.

5 The Methodology is clear, but the research site and the samples need to be elaborated upon.

Response: Thanks for your suggestion. In the revised manuscript, we tried to present the research site and samples as specifically as possible (see Line 261-262).

The revised version is:

85 Chinese universities/colleges were chosen as the research site. 

6 The Findings and Discussion summarize the results in clear and concise language. Please place your study within the context of previous studies more closely.

Response: We sincerely appreciate the valuable suggestion. We have also revised this part in a clearer way. We described more specifically of the implication part in the revised manuscript (See Line 451-499). Thank you again.

7 This paper features too many lengthy sentences. They not only make the paper unreadable and incomprehensible, but also lead to grammatical mistakes. Please do make every single sentence crystal clear and grammatically accurate first.

Response: We earnestly appreciate the valuable suggestion. We have tried our best to re-read and make some changes to the manuscript to improve the manuscript language. We shortened/removed some of the lengthy sentences to make them clearer. These changes will not influence the content and framework of the article. Here, we did not list the changes. Again, we appreciate for your warm work earnestly. (See #Editor, Q3)

Reviewer #2: 

1.Given the number of instructors included in the study, there were arguably few students included in comparison, I wonder why that was. I mean, there were too few students compared to the number of instructors.

Response: We sincerely appreciate the valuable comments. Data of the instructors and students were 1752 and 5914. We did our utmost to survey more of the instructors. Yet, it was a tough work to complete. This is one of the limitations of the current research and we will pay attention to such problems in the future researches. Again, thank you for your comments.

2.In page 9, the authors write: "In this adapted framework, the capitalized letters “I” and “S” refer to “instructor” and “student”". But I do not see these abbreviations being used in the literature review, perhaps these two paragraphs would be better located in the methodology? In any case, it is not evident why these two paragraphs are there. I think that this notation is not even used (or at least consistently used) when providing the results, conclusion or in the tables.

Response: We sincerely appreciate the valuable comments. We re-examined the notation here and removed them since they were not being used in the following work anymore (see Line 118-120). 

3.In page 11, it says "A total of 6024 students were withdrawn in the student questionnaire" withdrawn? Did you mean surveyed? Check the writing in this section, it is confusing.

Response: Thank you for figuring out the problem here. We change “withdrawn” to “surveyed”. Thank you again (See Line 258).

4.In page 12, what does "Subjective questions" mean?

Response: Thank you for figuring out the problem here. We are sorry to make you confused. In fact, it refers to some open-ended questions. Thus, we change it into “open-ended questions” in the revised manuscript (See Line 275).

5.The authors did not include enough information about the data collection instrument in the methods section. Also, the research design, specifically the pre-course and course delivery phases were not described.

Response: a. We sincerely appreciate the valuable comments. Data were collected through an online questionnaire, the website is https://www.wjx.cn/vm/tUrrbPw.aspx#. (See Line 243-245). The revised version is as follows:

This exploratory study was operated in inland China with two online questionnaires on the platform WenJuanxing (www.wjx.cn). It is a popular platform to collect questionnaire in China. Website of this research questionnaire is: https://www.wjx.cn/vm/tUrrbPw.aspx#. 

Response: b. More details were added in the revised manuscript:

The student e-readiness scale consists of 4 items for each of the three phases: pre-course, course delivery, and course completion phase (12 items in total), while the instructor e-readiness is composed of 5 items for the pre-course and course delivery phase respectively, and 6 items for the course completion phase (16 items in total) (See Line 2549-253). 

6.Formal results and discussion sections were not provided, the results are provided just below methodology without another main heading. I think that results may be presented in better manner and properly discussed.

Response: We sincerely appreciate the valuable comments and suggestion. We make some changes to the main heading, as “Results” and “Discussion” in the revised manuscript to meet the approval. 

7.Writing style should be improved, the manuscript is readable, but in some sections, it becomes difficult to understand what the authors are referring to. There are also minor grammar issues such as use of verb tenses, plural/singular, and use of articles.

Response: We tried our best to improve the manuscript and made some changes to the manuscript. These changes will not influence the content and framework of the paper. We appreciate for your warm work earnestly. (see Reviewer 1, Q7)

8.The authors have declared that data are fully available but then they stated that "All relevant data are within the manuscript and its Supporting Information files", I believe that given PLOS best practices, data should be fully provided in a data repository. Tables are valuable but they are not enough to suggest that all data were made available.

Response: We sincerely appreciate your valuable suggestion. We have put all the collected data in the revised manuscript. 

Reviewer #3: Thank you for submitting your research paper for PLOS. This paper covers an important topic. While it is a well-written paper, there are some issues that need to be addressed.

1. The novelty of this paper is not clear as many research papers have been published in this area. Thus, the novelty of this paper should be further justified by highlighting main contributions to the existing literature in order to be published in a reputable journals that seek new knowledge and insights.

Response: We sincerely appreciate your valuable suggestion. We revised the manuscript to meet the requirements (See Line 488-494). The revised version is as follows:

In contrast to prior studies, this study designed a questionnaire to collect data from 85 Chinese universities. Questionnaire were collected in more than 30 provinces and the sample size were virtually equally distribute in each province. Furthermore, this study followed the recommendations of Gay and Zou , who pointed out that items like “desk-help service” [1] and “prior online learning experiences” [5] should be covered in the future research. Moreover, the questionnaire covered more participants, making the research data more convincing.

2. The authors need to add a paragraph at the end of the introduction to show the structure of the paper.

Response: We sincerely appreciate your valuable suggestion. Based on the suggestion, we explained the structure of the paper at the end of the introduction (see Line 90-92).

3. The authors are required to provide a description of the data analysis methodology at the beginning of the data analysis section.

Response: We sincerely appreciate your valuable suggestion. We provided more information of the data analysis methodology to make it more specific (See Line 260). It is as follows:

Data were analyzed with the software SPSS.

4. The authors need to define the population of the study and sampling technique adopted. Also please justify why your chosen sampling technique and sample size are appropriate.

Response: a. We sincerely appreciate your valuable suggestion. Participants surveyed were mainly undergraduates and English teachers. Students were still having online English classes and teachers were teaching college English courses through online platforms (See Line 256-257). It is as follows:

Students were undergraduates who were having online college classes, and instructors were teachers teaching online college English in the EFL context. 

Response: b. Design for sample size needs to be perfected in the future design. This is one of the limitations of the current research (see Reviewer 2, Q1). Thank you again. 

5. Theoretical and practical implications should be reported after the discussion section. It is recommended to rely on the findings to provide thorough implications.

Response: We sincerely appreciate your valuable suggestion. We described more specifically of the implication part in the revised manuscript (See Line 470-499). Thank you again. 

6. The authors are advised to strengthen the literature review and discussion sections by including more recent and relevant literature. This includes but not limited to:

- Novel extension of the UTAUT model to understand continued usage intention of learning management systems: the role of learning tradition. Doi: https://doi.org/10.1007/s10639-021-10758-y

- Towards a Sustainable Adoption of E-Learning Systems: The Role of Self-Directed Learning. Doi: https://doi.org/10.28945/4980

- Developing a Holistic Success Model for Sustainable E-Learning: A Structural Equation Modeling Approach. Doi: https://doi.org/10.3390/su13169453

- Evaluating E-learning systems success: An empirical study. Doi: https://doi.org/10.1016/j.chb.2019.08.004

Response: We sincerely appreciate the suggestion. The recommended references are valuable and helpful for the research and we have added these literatures while revising the manuscript according to the suggestion (See Line 133-161).

---

## [Decision Letter · Decision Letter 1]

28 Mar 2023

Predicting student and instructor e-readiness and promoting e-learning success in online EFL class during the COVID-19 pandemic: A case from China

PONE-D-23-00431R1

Dear Dr. Yang,

We’re pleased to inform you that your manuscript has been judged scientifically suitable for publication and will be formally accepted for publication once it meets all outstanding technical requirements.

Kind regards,

Ehsan Namaziandost

Academic Editor

PLOS ONE

Additional Editor Comments (optional):

Dear authors,

Considering the comments, the paper has improved significantly, however, it is important follow the referencing style (in-text and reference list) followed by the journal.

Reviewers' comments:

Reviewer's Responses to Questions

**Comments to the Author**

1. If the authors have adequately addressed your comments raised in a previous round of review and you feel that this manuscript is now acceptable for publication, you may indicate that here to bypass the “Comments to the Author” section, enter your conflict of interest statement in the “Confidential to Editor” section, and submit your "Accept" recommendation.

Reviewer #1: All comments have been addressed

Reviewer #3: All comments have been addressed

2. Is the manuscript technically sound, and do the data support the conclusions?

Reviewer #1: Yes

Reviewer #3: Yes

3. Has the statistical analysis been performed appropriately and rigorously? 

Reviewer #1: Yes

Reviewer #3: Yes

4. Have the authors made all data underlying the findings in their manuscript fully available?

Reviewer #1: Yes

Reviewer #3: Yes

5. Is the manuscript presented in an intelligible fashion and written in standard English?

Reviewer #1: No

Reviewer #3: Yes

6. Review Comments to the Author

Reviewer #1: Dear authors,

I am happy with your revised version. I suggest that you should revise your language from A to Z.

Best wishes

Reviewer #3: Thank you for resubmitting you revised version. The paper has improved significantly after addressing the reviews comments. However, it is important follow the referencing style (in-text and reference list) followed by the journal.

7. PLOS authors have the option to publish the peer review history of their article (what does this mean?). If published, this will include your full peer review and any attached files.

Reviewer #1: **Yes: **Yongliang Wang Nanjing Normal University

Reviewer #3: **Yes: **Ahmad Samed Al-Adwan

---

## [Editor Report · Acceptance letter]

27 Apr 2023

PONE-D-23-00431R1 

Predicting student and instructor e-readiness and promoting e-learning success in online EFL class during the COVID-19 pandemic: A case from China 

Dear Dr. Yang:

I'm pleased to inform you that your manuscript has been deemed suitable for publication in PLOS ONE. Congratulations! Your manuscript is now with our production department. 

Kind regards, 

on behalf of

Dr. Ehsan Namaziandost 

Academic Editor

PLOS ONE